# Effects of Self-Esteem on the Association between Negative Life Events and Suicidal Ideation in Adolescents

**DOI:** 10.3390/ijerph16162846

**Published:** 2019-08-09

**Authors:** Yuhui Wan, Ruoling Chen, Shanshan Wang, Sophie Orton, Danni Wang, Shichen Zhang, Ying Sun, Fangbiao Tao

**Affiliations:** 1Department of Maternal, Child & Adolescent Health, School of Public Health, Anhui Medical University, Hefei 230032, China; 2Anhui Provincial Key Laboratory of Population Health & Aristogenics, Hefei 230032, China; 3Centre for Health and Social Care Improvement (CHSCI), Faculty of Education, Health and Wellbeing, University of Wolverhampton, Wolverhampton WV1 1LY, UK; 4Division of Primary Care, School of Medicine, University of Nottingham, Nottingham NG1 4BU, UK; 5Teaching Centre of Preventive Medicine, School of Public Health, Anhui Medical University, Hefei 230032, China

**Keywords:** negative life events, suicidal ideation, self-esteem, adolescents

## Abstract

Negative life events (NLEs) increase the risk of suicidal ideation (SI) in adolescents. However, it is not known whether the association between NLEs and SI can be moderated by self-esteem and varies with gender. The aim of the current paper was to examine gender differences in the association of SI with NLEs in adolescents, and assess the effects of self-esteem on the association and their gender variations. We conducted a school-based health survey in 15 schools in China between November 2013 and January 2014. A total of 9704 participants aged 11–19 years had sociodemographic data reported and self-esteem (Rosenberg self-esteem scale), NLEs, and SI measured. Multivariate-adjusted logistic regression was used to calculate the odds ratio (OR) of having SI in relation to NLEs. Increased risk of SI was significantly associated with NLEs (adjusted OR 2.19, 95%CI 1.94–2.47), showing no gender differences (in females 2.38, 2.02–2.80, in males 1.96, 1.64–2.36, respectively). The association was stronger in adolescents with high esteem (2.93, 2.34–3.68) than those with low esteem (2.00, 1.65–2.42) (ORs ratio 1.47, *p* = 0.012). The matched figures in females were 3.66 (2.69–4.99) and 2.08 (1.61–2.70) (1.76, *p* = 0.006), while in males these figures were 2.27(1.62–3.19) and 1.89 (1.41–2.53) (1.20, *p* = 0.422), respectively. Self-esteem had moderate effects on the association between NLEs and SI in adolescents, mainly in females. NLEs, self-esteem, and gender need to be incorporated into future intervention programs to prevent SI in adolescents.

## 1. Introduction

Suicidal ideation (SI) is a major public health problem in China and around the world [1,2]. Suicidal ideation is common among adolescents, present in about 17.5% of the middle school students in China [1] and about 17.7% of adolescents in the United States [3]. Suicidal ideation has been found to be significantly associated with suicidal attempt, depressiveness, anxiety, disordered eating, and so on [4,5,6,7]. To sustain improvements in management and prevention initiatives, we need to identify and understand the risk and protective factors associated with suicidal ideation.

Recently some authors have considered negative life events (NLEs) and self-esteem as an impact factors for SI in adolescents [8,9,10,11,12,13]. The direction of the relationship is that an increased perception in the number and seriousness of negative life events can increase SI. A variable that could be related to NLEs and SI is self-esteem. While negative life events can reduce self-esteem, people with high self-esteem have been found to buffer negative life events. However, not surprisingly, NLEs and self-esteem are highly correlated [14]. However, there is little research on the interaction effects between NLEs and self-esteem on SI in adolescents. A further factor that could be influential in this relationship is gender. Evidence differentiates males and females in terms of the prevalence and impact of different NLEs [15], perceptions of self-esteem [16], and the presentation of SI [17]. However, no study that the authors are aware of has examined gender differences in the interaction between NLEs and self-esteem on SI in adolescents. Greater understanding of the association between self-esteem, NLEs, and SI, and any gender differences between these, could inform the development of future interventions to prevent suicide attempts.

In this paper, we examine a large-scale health survey data to investigate the independent association of SI with NLEs in adolescents, the potential moderating effects of self-esteem on the association, and their gender differences. Based on the cited literature and Chinese context, we hypothesized that SI was positively associated with NLEs in adolescents. Furthermore, we hypothesized that self-esteem moderated the association, and their gender differences was obvious.

## 2. Methods

### 2.1. Sample and Procedures

This questionnaire survey was conducted in two cities (Zhengzhou in Henan province, and Guiyang in Guizhou province) from China, between November 2013 to January 2014. Eight junior and senior schools (four rural and four urban) were randomly selected from each city for the study, taking into consideration the representativeness of adolescent students and facilitation of data collection. As one school combined with junior and senior schools, 15 schools were selected for the survey. A total of 10,100 adolescents, with a range of 11 to 19 years, from grades 7–12 were recruited to participate in the study and asked to complete an anonymous questionnaire. Of them, 396 (3.9%) were excluded from the study because of unwillingness to respond to the questionnaire, high levels of missing data, or obviously fictitious responses. Thus, the data from 9704 participants were analyzed. Informed consent was sought from parents/guardians of each student prior to completion. The design and data collection procedures were approved by the Ethics Committee of Anhui Medical University (2012534).

### 2.2. Data Collection

#### 2.2.1. Demographic and Control Variables

Demographic data for each participant was recorded, including age, gender (males or females), urban/rurality, only child (yes or no), parents’ education level (less than junior middle school, junior middle school, senior middle school, college or more) and economic status of the family (poor, moderate, or good).

Psychological symptoms and adverse childhood experiences (ACEs) were collected and included as confounders for adjustment in the final model analysis. Psychological symptoms were evaluated in 39 items, including emotional symptoms, conduct symptoms, and social adaptation symptoms, which were validated by psychological domain of the “Multidimensional Sub-health Questionnaire of Adolescents” (MSQA) [18] (Cronbach’s α = 0.954 in the present study). In accordance with the national norm established for the MSQA in China [18], the 90th percentile of national norm was selected as the cut-off point. Psychological symptoms were treated as dichotomous variables.

ACEs were defined as having experienced childhood maltreatment and/or household dysfunction. Childhood maltreatment was evaluated using the Child Trauma Questionnaire (CTQ) [19], which assesses five different forms of childhood trauma (physical abuse, sexual abuse, emotional abuse, physical neglect, and emotional neglect). Household dysfunction questions were derived from the version of Centers for Disease Control (CDC) and Kaiser Permanente Adverse Childhood Experiences Study in America [20]. In the present study, Cronbach’s alpha coefficient for the ACEs scale was 0.758. Due to the high inter-relatedness of various types of adverse childhood experiences (all *p* < 0.01), the “number of different types of ACEs” score was created by summing the dichotomous ACEs items (range: 0 (unexposed) to 6 (exposed to physical abuse, sexual abuse, emotional abuse, physical neglect, emotional neglect and household dysfunction)) [21]. The scores of ACEs were treated as none ACEs (0), low ACEs (1–2), and high ACEs (≥3) in the current study.

#### 2.2.2. Suicidal Ideation

Suicidal ideation was defined as a “yes” in response to the question “In the past 12 months, have you ever thought about killing yourself?”, as referred to in the “middle school questionnaire” of the 2013 Youth Risk Behavior Surveillance System (YRBSS) in the United States [22].

#### 2.2.3. Negative Life Events

Negative life events were examined by the multidimensional life events rating questionnaire of middle school students in China [23]. The 43-item questionnaire includes several domains: family, school, partner, health, and love. For each item, adolescents indicated whether each event had occurred during the past 6 months; if yes, they then indicated the impact of each event on them, using a 5-point (0–4 score) scale ranging from “none” to “extremely severe”. Items which had not occurred were scored as “0”. In accordance with the national norm established for the multidimensional life events rating questionnaire in China [23], negative life events were treated as dichotomous variables (≥28 high or <28 low).

#### 2.2.4. Self-Esteem

Self-esteem was assessed using Rosenberg self-esteem scale [24]. The scale is a 10-item measure that assesses adolescents’ global feelings of self-worth or value. The items are presented in a 4-point Likert format ranging from “strongly agree” to “strongly disagree”. Half of the items are positively worded, and others are negatively worded. The Rosenberg self-esteem scale is a widely used measure to evaluate self-esteem and has been shown to have good reliability [24]. The scale scores, with a possible range from 0 to 30 (low to high self-esteem), had an accepted internal consistency (Cronbach alpha) of 0.765 in the current study. To facilitate interpretation, the total scores were then converted into tertiles (low, moderate and high from low to high scores).

### 2.3. Statistical Analysis

Differences in socio-demographic risk factors, psychological symptoms and self-esteem between the SI and non-SI groups were assessed using chi-squared tests for categorical variables and one-way analysis of variance for continuous variables. Binomial logistic regression models were used to examine the associations between NLEs and SI in various self-esteem groups. In the models, age, gender, urban/rurality, only child status, parents’ education level, economic status of family, and psychological symptoms were adjusted for.

We tested differences in the association of NLEs with SI between males and females and among different levels of self-esteem, using a 2-sided *p*-value and calculated a ratio of two relative risks as we did before [25]. All analyses were conducted with SPSS software, version 16.0 (SPSS Inc., Chicago, IL, USA).

## 3. Results

The mean age of the 9704 participating students was 15.6 years (SD = 1.8), and 5104 participants (52.6%) were female. A total of 1791 students (18.5%) reported having SI in the past year. The characteristics of participants with and without SI are shown in Table 1. Those adolescents with SI were more likely to be younger students, females, from urban area, only child, and to have good or poor family economic status, higher parents’ education level, higher ACEs, psychological symptoms and low self-esteem.

The association between NLEs and SI was significant, with adjustment for other factors, including self-esteem (Table 2). This association remained in males and females when analyzed separately, without gender differences in these ORs. In the model, when we tested the interaction effect between NLEs and self-esteem on SI, it was significant for total sample (*p* = 0.014) and for females (*p* = 0.012), but not for males (*p* = 0.552).

Table 3 shows the association of SI with NLEs in the different levels of self-esteem. Adolescents with high level of self-esteem had a higher risk of SI in relation to NLEs than those with the moderate and low self-esteem (corresponding ratio of two odds ratios (RORs) 1.51, 1.10–2.06 and 1.47, 1.09–1.97 respectively, *p* < 0.05). These differences remained significant in females (Table 4) (corresponding RORs 1.86, 1.22–2.82 and 1.76, 1.18–2.63 respectively, *p* < 0.05), but not in males (Table 5). On examining gender differences in the association of SI with NLEs by different levels of self-esteem, the association was found to be stronger in females with high self-esteem than in males (RORs 1.61, 1.02–2.55, *p* = 0.041), but in those with moderate or low self-esteem there were no statistical differences between females and males (corresponding RORs 1.08 (0.70–1.66) and 1.10 (0.72–1.67) respectively, *p* > 0.05).

## 4. Discussion

Previous findings [26,27] indicated that individuals with high level NLEs increased the risk of SI independently, which is consistent with those in the present study. A prospective study of American adolescents and young adults reported that exposure to major loss life events were related to subsequent increases in SI [26]. However, in a study of Chinese rural adolescents a clear dose–response relationship between the number of NLEs and SI was initially found, but the significant association disappeared after controlling for internalizing and externalizing problems [28]. In the current study, an independent association between NLEs and SI was found in both males and females, and there were no significant gender differences in the effect. We have noted that some studies investigated the gender differences in the relationship [29,30]. In South Korea a follow-up study of the 8th grade cohort found that adverse life events predicted the incidence of SI in male but not in female students [29]. However, a nationwide school-based mental health screening test data in South Korea showed that adverse life events (e.g., family conflict, violence and bullying) significantly contributed to suicidal ideation in both genders [30]. The gender differences in the impact of NLEs on SI require further investigation.

High self-esteem has been suggested as a factor of “resilience”, whereas low self-esteem may sensitize the individual to make negative appraisals when exposed to stressful life events [31]. Few studies have been done to examine the moderate effect of self-esteem on the association of NLEs and SI in adolescents or young adults with inconsistent results [32,33]. A survey in Taiwanese adolescents (aged 15–19 years) found that high self-esteem was an important protective factor for SI, but self-esteem did not moderate the effect of life stress on SI [33]. In the United States, Wilburn et al. carried out a study of college students and found no significant interaction effects between stress life events and self-esteem on SI [31]. However, a further study conducted in the USA in undergraduates showed that greater satisfaction of self-competence could provide protection from SI and help buffer the effect of NLEs [34]. Contrary to the findings of above research, our current study found the association between NLEs and SI was stronger in adolescents with high self-esteem than those with moderate or low esteem, suggesting that high self-esteem among those who have accumulated NLEs could be associated with increased risk of SI. High self-esteem may allow a person to interact more effectively with their environment, and thus be less likely to be exposed to a high level of NLEs or interpret negative events more positively [14]. However, the findings of our study indicated that high self-esteem might have harmful effects. The mechanisms behind this are unclear. Possible explanations for this association are that individuals with very high self-esteem may consider that they are incapable of failing and indicative of perfectionist, a concept more readily linked to suicide thoughts [8]. When encountering NLEs, people with high levels of self-esteem often cannot accept their own failures, and lack of effective response experience may be more likely to produce suicidal ideation [35]. Further research is needed to better understand the role of self-esteem in the relationship between NLEs and SI.

The current study has added to the existing literature by demonstrating gender differences in the association of SI with NLEs. A possible explanation for the observed gender differences could be due to differences in seeking help; males may have a tendency to resolve their problems on their own and consider seeking help as incompetency, whereas females are less likely to consider the need for help as a negative and may be more willing to ask for help [36]. In the Chinese traditional culture males are more likely to be socialized to be independent and females are more likely to seek help from outside when having trouble. Thus females, especially for those with high self-esteem, maybe more sensitive to NLEs and interpersonal pressure [37]. This finding may be explained in the Chinese cultural context. In China, the genders have different role expectations. Compared to females, males are expected to undertake more responsibilities and pressures from family and society [38]. Therefore, boys have more experience and strategies than girls to deal with NLEs, such as conflicting with family members and friends, which in turn leads to suicidal ideation. These findings could help us identify adolescents who are most at risk of SI, and develop psychological interventions that take into consideration self-esteem among females with high NLEs.

### Strength and Limitation

The main strength of this study is its novelty, examining the potentially important association between NLE, self-esteem, gender, and SI in adolescents, which, to the author’s knowledge has not been previously explored.

The study sample is large, covering urban and rural areas, and has provided enough statistical power to examine gender differences, with multivariate adjustment analysis. However, several limitations should be considered when interpreting these results. First, the sample was selected by convenience from Guiyang and Zhengzhou city, which limits the generalizability of the results to other areas. Further studies in more heterogeneous and national adolescent populations are needed to confirm the findings. Second, our study is cross-sectional design, precluding an exploration of the causal associations between NLEs, self-esteem and SI. However, previous cohort studies [39,40] showed that high NLEs and low self-esteem increased the risk of SI, which were similar to our findings. Third, due to the reliance on self-reported questionnaires we could not exclude potential recall bias and rates of NLEs and SI may therefore be under-reported due to the sensitive nature of the questions.

## 5. Conclusions

NLEs are independently associated with an increased risk of SI in both female and male adolescents. The association was stronger in adolescents with high self-esteem than those who had moderate or low esteem, and it is particularly seen in females. These findings have implications for the development of future interventions to reduce SI in adolescents. For example, the development of appropriate self-esteem with mindfulness training in the high-risk adolescent population, involving body scan, breath meditation, and emotion and thought meditation, could prevent an expected increase in SI [41]. Beyond education and practical assistance, peer mentoring or support groups may be effective in reducing vulnerability of students to negative life events by increasing social support [42]. In addition, attention should also be paid to the differences between students of different genders in responding to self-esteem and negative life events.

## Figures and Tables

**Table 1 ijerph-16-02846-t001:** Characteristics of participants by SI, *n* (%): 15-school study, China.

Variables	Total *n* = 9704	No SI *n* = 7913	SI *n* = 1791	*p*-Value
**Age (mean, SD)**	15.6 (1.8)	15.6 (1.8)	15.4 (1.7)	<0.001
**Gender**				
Female	5104 (52.6)	4057 (79.5)	1047 (20.5)	<0.001
Male	4600 (47.4)	3856 (83.8)	744 (16.2)	
**Urban/rurality**				
urban	4049 (41.7)	3160 (78.0)	889 (22.0)	<0.001
rural	5655 (58.3)	4753 (84.0)	902 (16.0)	
**Only child**				
yes	2999 (30.9)	2385 (79.5)	614 (20.5)	0.001
no	6705 (69.1)	5528 (82.4)	1177 (17.6)	
**Father’s education level**				
less than junior middle school	2256 (23.2)	1889 (83.7)	367 (16.3)	<0.001
junior middle school	3802 (39.2)	3150 (82.9)	652 (17.1)	
senior middle school	2107 (21.7)	1652 (78.4)	455 (21.6)	
college or more	1539 (15.9)	1222 (79.4)	317 (20.6)	
**Mother’s education level**				
less than junior middle school	3096 (31.9)	2574 (83.1)	522 (16.9)	<0.001
junior middle school	3478 (35.8)	2863 (82.3)	615 (17.7)	
senior middle school	1888 (19.5)	1501 (79.5)	387 (20.5)	
college or more	1242 (12.8)	975 (78.5)	267 (21.5)	
**Economic status of family**				
good	1289 (13.3)	1022 (79.3)	267 (20.7)	0.031
moderate	6638 (68.4)	5456 (82.8)	1182 (17.8)	
poor	1777 (18.3)	1435 (80.8)	342 (19.2)	
**Psychological symptoms**				
no	7414 (76.4)	6472 (87.3)	942 (12.7)	<0.001
yes	2290 (23.6)	1441 (62.9)	849 (37.1)	
**ACEs**				
0	1243 (12.8)	1140 (91.7)	103 (8.3)	<0.001
1–2	4385 (45.2)	3755 (85.6)	630 (14.4)	
≥3	4076 (42.0)	3018 (74.0)	1058 (26.0)	
**Self-esteem**				
high	3513 (36.2)	3062 (87.2)	451 (12.8)	<0.001
moderate	3261 (33.6)	2755 (84.5)	506 (15.5)	
low	2930 (30.2)	2096 (71.5)	834 (28.3)	

Abbreviations: SI = suicidal ideation; ACEs = adverse childhood experiences.

**Table 2 ijerph-16-02846-t002:** Number, % and adjusted odds ratio (OR) of SI by level of NLE: 15-school study, China.

NLEs	SI, *n* (%)			Adjusted Analysis ^a^		Adjusted Analysis ^b^		Adjusted Analysis ^c^	
	No	Yes	*p*-Value	OR (95%CI)	*p*-Value	OR (95%CI)	*p*-Value	OR (95%CI)	*p*-Value
Total									
low	5186 (88.7)	659 (11.3)	<0.001	1.0		1.0		1.0	
high	2727 (70.7)	1132 (29.3)		3.39 (3.04–3.78)	<0.001	2.29 (2.03–2.58)	<0.001	2.19 (1.94–2.47)	<0.001
Female									
low	2659 (87.8)	369 (12.2)	<0.001	1.0		1.0		1.0	
high	1398 (67.3)	678 (32.7)		3.64 (3.16–4.21)	<0.001	2.49 (2.12–2.92)	<0.001	2.38 (2.02–2.80)	<0.001
Male									
low	2527 (89.7)	290 (10.3)	<0.001	1.0		1.0		1.0	
high	1329 (74.5)	454 (25.5)		3.09 (2.62–3.63)	<0.001	2.05 (1.71–2.46)	<0.001	1.96 (1.64–2.36)	<0.001

Abbreviations: SI = suicidal ideation; NLEs = negative life events. ^a^ Adjusted for gender, age. ^b^ Adjusted for gender, age, urban/rurality, only child, parents’ education level, economic status of family, psychological symptoms, ACEs. ^c^ Adjusted for gender, age, urban/rurality, only child, parents’ education level, economic status of family, psychological symptoms, ACEs, self-esteem.

**Table 3 ijerph-16-02846-t003:** Number, % and OR of SI in relation to NLEs by level of self-esteem: 15-school study, China.

Self-Esteem	NLEs	*n* (%)	Adjusted Analysis ^a^		Adjusted Analysis ^b^	
			OR (95%CI)	*p*-Value	OR (95%CI)	*p*-Value
high	low	205 (8.1)	1.0		1.0 ^a^	
	high	246 (25.4)	4.09 (3.33–5.02)	<0.001	2.93 (2.34–3.68)	<0.001
moderate	low	212 (10.8)	1.0		1.0 ^a^	
	high	294 (22.8)	2.55 (2.10–3.11)	<0.001	1.94 (1.56–2.40)	<0.001
low	low	242 (18.2)	1.0		1.0 ^a^	
	high	592 (37.0)	2.72 (2.28–3.23)	<0.001	2.00 (1.65–2.42)	<0.001
Ratio of two adjusted ^b^ ORs in high vs. moderate self-esteem					1.51 (1.10–2.06)	0.010
Ratio of two adjusted ^b^ ORs in high vs. low self-esteem					1.47 (1.09–1.97)	0.012

Abbreviations: SI = suicidal ideation; NLEs = negative life events. ^a^ Adjusted for age, gender. ^b^ Adjusted for age, urban/rurality, only child, parents’ education level, economic status of family, psychological symptoms, ACEs.

**Table 4 ijerph-16-02846-t004:** Number, % and OR of SI in relation to NLEs by level of self-esteem in females: 15-school study, China.

Self-Esteem	NLEs	*n* (%)	Adjusted Analysis ^a^		Adjusted Analysis ^b^	
			OR (95%CI)	*p*-Value	OR (95%CI)	*p*-Value
high	low	109 (8.8)	1.0		1.0	
	high	149 (31.5)	4.91 (3.72–6.49)	<0.001	3.66 (2.69–4.99)	<0.001
moderate	low	127 (11.6)	1.0		1.0	
	high	172 (24.8)	2.63 (2.03–3.39)	<0.001	1.97 (1.49–2.61)	<0.001
low	low	133 (19.1)	1.0		1.0	
	high	357 (39.3)	2.86 (2.26–3.62)	<0.001	2.08 (1.61–2.70)	<0.001
Ratio of two adjusted ^b^ ORs in high vs. moderate self-esteem					1.86 (1.22–2.82)	0.004
Ratio of two adjusted ^b^ ORs in high vs. low self-esteem					1.76 (1.18–2.63)	0.006

Abbreviations: SI = suicidal ideation; NLEs = negative life events. ^a^ Adjusted for age. ^b^ Adjusted for age, urban/rurality, only child, parents’ education level, economic status of family, psychological symptoms, ACEs.

**Table 5 ijerph-16-02846-t005:** Number, %, and OR of SI in relation to NLEs by level of self-esteem in males: 15-school study, China.

Self-Esteem	NLEs	*n* (%)	Adjusted Analysis ^a^		Adjusted Analysis ^b^	
			OR (95%CI)	*p*-Value	OR (95%CI)	*p*-Value
high	low	96 (7.3)	1.0		1.0	
	high	97 (19.6)	3.27 (2.40–4.45)	<0.001	2.27 (1.62–3.19)	<0.001
moderate	low	85 (9.7)	1.0		1.0	
	high	122 (20.4)	2.46 (1.82–3.33)	<0.001	1.83 (1.32–2.55)	<0.001
low	low	109 (17.2)	1.0		1.0	
	high	235 (34.0)	2.56 (1.97–3.32)	<0.001	1.89 (1.41–2.53)	<0.001
Ratio of two adjusted ^b^ ORs in high vs. moderate self-esteem					1.24 (0.77–1.99)	0.371
Ratio of two adjusted ^b^ ORs in high vs. low self-esteem					1.20 (0.77–1.88)	0.422

Abbreviations: SI = suicidal ideation; NLEs = negative life events. ^a^ Adjusted for age. ^b^ Adjusted for age, urban/rurality, only child, parents’ education level, economic status of family, psychological symptoms, ACEs.

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
