# Peer review of "Effects of Self-Esteem on the Association between Negative Life Events and Suicidal Ideation in Adolescents"

_ijerph, 2019, doi:10.3390/ijerph16162846_

Round 1
Reviewer 1 Report
This article fails to specify the cultural context of the research. China (PRC) has a unique profile in world comparisons, having high rates of completed suicide in both males and females. This could be an artifact of legal and forensic systems for determining cause of death, but this needs to be discussed. The implicit psychosocial model of social factors influencing suicidal ideas, depression, self-esteem and suicide is naive and assumes that models from, say North America, pertain in PRC. There is no awareness of the evidence from cross-cultural psychology on the need for back-translation of psychological instruments, and estimates of their validity and reliability in a new culture, following translation. In our own work with the Rosenberg Self-Esteem Scale (RSES) in Hong Kong, Japan and England we found that the "cultural meanings" of the scale were embedded in a particular culture, and causal models trying to predict suicidality had to be culturally specific.
The problems of trying to predict the risk of completed suicide (a rare event) from psychological impairment on psychosocial measures are not addressed adequately. The best predictor of a potentially lethal suicide attempt is a non-lethal suicide attempt. Suicidal ideation is frequent and is a very poor predictor of completed suicide. The variable of psychological resilience needs to be measured in a new longitudinal study. The correlations presented in this article do not, in my opinion, have any public health implications. If self-esteem is important how can interventions (eg by school counsellors) improve self-esteem and reduce suicidal risk?
The article could be rewritten as guiding a set of hypotheses for a new longitudinal research programme. Recent literature needs to be incorporated into this revision, including the following:
Perrot, C., Vera, L., & Gorwood, P. (2018). Poor self-esteem is correlated with suicide intent, independently from the severity of depression. L'Encephale, 44(2), 122-127.
Montes-Hidalgo, J., & Tomás-Sábado, J. (2016). Self-esteem, resilience, locus of control and suicide risk in nursing students. Enfermeria clinica, 26(3), 188-193.
Lehmann, M., Hilimire, M. R., Yang, L. H., Link, B. G., & DeVylder, J. E. (2016). Investigating the relationship between self-esteem and stigma among young adults with history of suicide attempts. Crisis.
Fiorilli, C., Capitello, T. G., Barni, D., Buonomo, I., & Gentile, S. (2019). Predicting adolescent depression: The interrelated roles of self-esteem and interpersonal stressors. Frontiers in psychology, 10.
Case, J. A., Burke, T. A., Siegel, D. M., Piccirillo, M. L., Alloy, L. B., & Olino, T. M. (2019). Functions of non-suicidal self-injury in late adolescence: a latent class analysis. Archives of suicide research, 1-22.
Lu, R., Zhou, Y., Wu, Q., Peng, X., Dong, J., Zhu, Z., & Xu, W. (2019). The effects of mindfulness training on suicide ideation among left‐behind children in China: A randomized controlled trial. Child: care, health and development, 45(3), 371-379.
Choi, Y. S., Shin, H. K., Hong, D. Y., Kim, J. R., Kang, Y. S., Jeong, B., ... & Lee, K. H. (2019). Self-esteem as a Moderator of the Effects of Happiness, Depression, and Hostility on Suicidality Among Early Adolescents in Korea. Journal of Preventive Medicine and Public Health, 52(1), 30.
Perrot, C., Vera, L., & Gorwood, P. (2018). Poor self-esteem is correlated with suicide intent, independently from the severity of depression. L'Encephale, 44(2), 122-127.
Montes-Hidalgo, J., & Tomás-Sábado, J. (2016). Self-esteem, resilience, locus of control and suicide risk in nursing students. Enfermeria clinica, 26(3), 188-193.
Lehmann, M., Hilimire, M. R., Yang, L. H., Link, B. G., & DeVylder, J. E. (2016). Investigating the relationship between self-esteem and stigma among young adults with history of suicide attempts. Crisis.
Fiorilli, C., Capitello, T. G., Barni, D., Buonomo, I., & Gentile, S. (2019). Predicting adolescent depression: The interrelated roles of self-esteem and interpersonal stressors. Frontiers in psychology, 10.
Case, J. A., Burke, T. A., Siegel, D. M., Piccirillo, M. L., Alloy, L. B., & Olino, T. M. (2019). Functions of non-suicidal self-injury in late adolescence: a latent class analysis. Archives of suicide research, 1-22.
Lu, R., Zhou, Y., Wu, Q., Peng, X., Dong, J., Zhu, Z., & Xu, W. (2019). The effects of mindfulness training on suicide ideation among left‐behind children in China: A randomized controlled trial. Child: care, health and development, 45(3), 371-379.
Choi, Y. S., Shin, H. K., Hong, D. Y., Kim, J. R., Kang, Y. S., Jeong, B., ... & Lee, K. H. (2019). Self-esteem as a Moderator of the Effects of Happiness, Depression, and Hostility on Suicidality Among Early Adolescents in Korea. Journal of Preventive Medicine and Public Health, 52(1), 30.
Author Response
Comments: This article fails to specify the cultural context of the research. China (PRC) has a unique profile in world comparisons, having high rates of completed suicide in both males and females. This could be an artifact of legal and forensic systems for determining cause of death, but this needs to be discussed. The implicit psychosocial model of social factors influencing suicidal ideas, depression, self-esteem and suicide is naive and assumes that models from, say North America, pertain in PRC. There is no awareness of the evidence from cross-cultural psychology on the need for back-translation of psychological instruments, and estimates of their validity and reliability in a new culture, following translation. In our own work with the Rosenberg Self-Esteem Scale (RSES) in Hong Kong, Japan and England we found that the "cultural meanings" of the scale were embedded in a particular culture, and causal models trying to predict suicidality had to be culturally specific.
Response: Thanks for your suggestion. We have made some changes to the first paragraph of Introduction section - “Suicidal ideation is a major public health problem in China and around the world. It is common to have suicidal ideation among adolescents, accounting for about 17.5% of the middle school students in China and about 17.7% of adolescents in US. Suicidal ideation (SI) has been found to be significantly associated with suicidal attempt, depressiveness, anxiety, disordered eating, and so on. To sustain improvements in management and prevention initiatives, we need to identify and understand the risk and protective factors associated with suicidal ideation.” in the manuscript, page 1, lines 34-40.
The authors totally agree with the reviewer that cultural-specific problems exist in some international psychological instruments. In our study, self-esteem was assessed using the Rosenberg self-esteem scale, which has been used widely in China, and the Cronbach's α coefficient for the RSES was 0.765. In recent years, several studies have used Rosenberg self-esteem scale to evaluate the level of Chinese adolescents’ self-esteem, the Cronbach's α coefficient were 0.83 [1], 0.85 [2] and 0.816 [3], respectively. Related studies using the Rosenberg self-esteem scale to assess self-esteem have shown significant associations of self-esteem with risk-taking, suicidal ideation, and depression among Chinese adolescents [1-4].
Comments: The problems of trying to predict the risk of completed suicide (a rare event) from psychological impairment on psychosocial measures are not addressed adequately. The best predictor of a potentially lethal suicide attempt is a non-lethal suicide attempt. Suicidal ideation is frequent and is a very poor predictor of completed suicide. The variable of psychological resilience needs to be measured in a new longitudinal study. The correlations presented in this article do not, in my opinion, have any public health implications. If self-esteem is important how can interventions (eg by school counsellors) improve self-esteem and reduce suicidal risk?
Response: We thank the reviewer for raising such an important direction for our future study. A longitudinal study is in great need to explore whether self-esteem and NLEs have predictive effects on suicidal ideation.
To develop appropriate self-esteem level through mindfulness training in the high-risk adolescent population could prevent an expected increase in SI [5]. Exercises included mindfulness meditation practice (body scan, breath meditation, emotion and thought meditation...) as well as mindfulness skills (Recognize, Allow, Investigate, Non-identify (RAIN) technique...) [6-8].
Comments: The article could be rewritten as guiding a set of hypotheses for a new longitudinal research programme. Recent literature needs to be incorporated into this revision, including the following:
Perrot, C., Vera, L., & Gorwood, P. (2018). Poor self-esteem is correlated with suicide intent, independently from the severity of depression. L'Encephale, 44(2), 122-127.
Montes-Hidalgo, J., & Tomás-Sábado, J. (2016). Self-esteem, resilience, locus of control and suicide risk in nursing students. Enfermeria clinica, 26(3), 188-193.
Lehmann, M., Hilimire, M. R., Yang, L. H., Link, B. G., & DeVylder, J. E. (2016). Investigating the relationship between self-esteem and stigma among young adults with history of suicide attempts. Crisis.
Fiorilli, C., Capitello, T. G., Barni, D., Buonomo, I., & Gentile, S. (2019). Predicting adolescent depression: The interrelated roles of self-esteem and interpersonal stressors. Frontiers in psychology, 10.
Case, J. A., Burke, T. A., Siegel, D. M., Piccirillo, M. L., Alloy, L. B., & Olino, T. M. (2019). Functions of non-suicidal self-injury in late adolescence: a latent class analysis. Archives of suicide research, 1-22.
Lu, R., Zhou, Y., Wu, Q., Peng, X., Dong, J., Zhu, Z., & Xu, W. (2019). The effects of mindfulness training on suicide ideation among left‐behind children in China: A randomized controlled trial. Child: care, health and development, 45(3), 371-379.
Choi, Y. S., Shin, H. K., Hong, D. Y., Kim, J. R., Kang, Y. S., Jeong, B., ... & Lee, K. H. (2019). Self-esteem as a Moderator of the Effects of Happiness, Depression, and Hostility on Suicidality Among Early Adolescents in Korea. Journal of Preventive Medicine and Public Health, 52(1), 30.
Response: Thanks for recommending these important literatures. After careful reading, the authors have decided to put the articles of Perrot et al, Lehmann, Fiorilli et al, and Lu et al as references 12, 7, 13 and 41 respectively in the revised manuscript.
Tian L, Dong X, Xia D, Liu L, Wang D. Effect of peer presence on adolescents' risk-taking is moderated by individual self-esteem: An experimental study. Int J Psychol. 2019. https://doi.org/10.1002/ijop.12611. You Z, Zhang Y, Zhang L, XuY, Chen X.2019. How does self-esteem affect mobile phone addiction? The mediating role of social anxiety and interpersonal sensitivity. Psychiatry Res, 2019, 271: 526-531. https://doi.org/10.1016/j.psychres.2018.12.040. Gao F, Guo Z, Tian Y, Si Y, Wang P. Relationship between shyness and generalized pathological internet use among Chinese school students: The serial mediating roles of loneliness, depression, and self-esteem. Front Psychol, 2018, 9:1822. https://doi.org/10.3389/fpsyg.2018.01822. Zhu X, Tian L, Huebner ES. Trajectories of suicidal ideation from middle childhood to early adolescence: Risk and protective factors. J Youth Adolesc. 2019. https://doi.org/10.1007/s10964-019-01087-y. Lu R, Zhou Y, Wu Q, Peng X, Dong J, Zhu Z, Xu W. The effects of mindfulness training on suicide ideation among left‐behind children in China: A randomized controlled trial. Child: care, health and development, 2019, 45(3):371-379.
https://doi.org/10.1111/cch.12650.
Leary MR, Baumeister RF. The nature and function of self-esteem: sociometer theory. Adv Exp Soc Psychol, 2000, 32(1):1–62. Brown JD, Collins RL, Schmidt GW. Self-esteem and direct versus indirect forms of self-enhancement. J Pers Soc Psychol, 1988, 55(3):445-453. Rasmussen MK, Pidgeon AM. The direct and indirect benefits of dispositional mindfulness on self-esteem and social anxiety. Anxiety, Stress, & Coping, 2011, 24(2):227-233.
Reviewer 2 Report
This is, in summary, a paper aimed to investigate gender differences in the association of suicidal ideation (SI) with negative life events (NLEs) in adolescents, and assess the effects of self-esteem on the association and their gender variations according to a school-based health survey (9,704 participants) conducted in 15 Chinese schools between November 2013 to January 2014. The authors found that increased risk of SI was associated with NLEs, showing no gender differences. They also added that the association was stronger in adolescents with high esteem than those with low esteem. The matched figures in females were 3.66 and 2.08, while in males were 2.27 and 1.89. Finally, self-esteem had moderate effects on the association between NLEs and SI in adolescents, mainly in females.
The authors may find as follows my main comments/suggestions.
First, when throughout the Introduction section, the authors correctly reported that suicide is an important phenomenon associated with significant disability and psychosocial impairment, they might even mention the relation between depression, and suicidal risk which is enhanced in patients with medical disorders. Specifically, the existence of depression, prior and current history of medical disorders, and cognitive impairment were reported to be the most important risk factors for suicide (PMID: 25491561). In addition, the authors adequately referred to the link between stressfull adverse life events’ exposure and negative outcome. According to the main results of a systematic review, the number of the experienced adversities or negative life events seemed to have a positive dose-response relation with youth negative outcome (PMID: 26303813). Thus, in order to briefly address these two important topics (although i understand that the link between subjective well-being, depression, negative life events, and negative outcome is not the main topic of this paper), i suggest to cite within the main text first the paper published on Eur Child Adolesc Psychiatry in 2015 (PMID: 25491561) and later the systematic review published on Drugs Aging in 2015 (PMID: 25491561).
Moreover, as the authors reported extensively the most important aims/objectives of this paper, the main study hypotheses should be similarly described in a more detailed manner.
Furthermore, the main psychometric instruments of this study could be described more succinctly.
In addition, within the first lines of the Discussion section, the authors do not need to repeat again what are the most relevant aims of this paper with regard to the main topic, as these objectives have been already discussed extensively elsewhere. Here, i suggest to immediately focus on the most relevant findings of the study and their implications for the general readership.
Moreover, some statements and assumptions within the main text need further additional clarifications. For instance, why the association between increased SI risk and NLEs was stronger only in female adolescents with high self esteem than those with moderate or low esteem need to be further discussed based on the current version of this paper. Simatly when the authors reported that high self-esteem might have harmful effects but the mechanisms behind this are unclear, here more information are required.
Finally, the authors should more deeply discuss what are, according to their expertise, the most relevant future interventions in order to reduce SI and suicide attempts in adolescents. Here, more details and some conclusive remarks about this topic are needed.
Author Response
Comments: First, when throughout the Introduction section, the authors correctly reported that suicide is an important phenomenon associated with significant disability and psychosocial impairment, they might even mention the relation between depression, and suicidal risk which is enhanced in patients with medical disorders. Specifically, the existence of depression, prior and current history of medical disorders, and cognitive impairment were reported to be the most important risk factors for suicide (PMID: 25491561). In addition, the authors adequately referred to the link between stressfull adverse life events’ exposure and negative outcome. According to the main results of a systematic review, the number of the experienced adversities or negative life events seemed to have a positive dose-response relation with youth negative outcome (PMID: 26303813). Thus, in order to briefly address these two important topics (although i understand that the link between subjective well-being, depression, negative life events, and negative outcome is not the main topic of this paper), i suggest to cite within the main text first the paper published on Eur Child Adolesc Psychiatry in 2015 (PMID: 25491561) and later the systematic review published on Drugs Aging in 2015 (PMID: 25491561).
Response: In view of the comments made by another reviewer, we have changed the first paragraph of the Introduction section to “Suicidal ideation is a major public health problem in China and around the world. It is common to have suicidal ideation among adolescents, accounting for about 17.5% of the middle school students in China and about 17.7% of adolescents in US. Suicidal ideation (SI) has been found to be significantly associated with suicidal attempt, depressiveness, anxiety, disordered eating, and so on. To sustain improvements in management and prevention initiatives, we need to identify and understand the risk and protective factors associated with suicidal ideation.”, so after considering, we quoted the article with a PMID of 26380813 as reference 11.
Comments: Moreover, as the authors reported extensively the most important aims/objectives of this paper, the main study hypotheses should be similarly described in a more detailed manner.
Response: Thanks for your suggestion. We have added the main study hypotheses - “Based on the cited literature and Chinese context, we hypothesized that SI was positively associated with NLEs in adolescents. Furthermore, we hypothesized that self-esteem moderated the association, and their gender differences was obvious.” in the manuscript, page 2, lines 13-16.
Comments: Furthermore, the main psychometric instruments of this study could be described more succinctly.
Response: We have done these accordingly (see the revised manuscript, page 2, lines 37-48; page 3, lines 1-6.)
Comments: In addition, within the first lines of the Discussion section, the authors do not need to repeat again what are the most relevant aims of this paper with regard to the main topic, as these objectives have been already discussed extensively elsewhere. Here, i suggest to immediately focus on the most relevant findings of the study and their implications for the general readership.
Response: We agree with the reviewer, and have deleted the first paragraph of the Discussion section about the most relevant aims of this paper with regard to the main topic.
Comments: Moreover, some statements and assumptions within the main text need further additional clarifications. For instance, why the association between increased SI risk and NLEs was stronger only in female adolescents with high self esteem than those with moderate or low esteem need to be further discussed based on the current version of this paper. Simatly when the authors reported that high self-esteem might have harmful effects but the mechanisms behind this are unclear, here more information are required.
Response: We have provided additional clarifications - “Thus females, especially for those with high self-esteem, maybe more sensitive to NLEs and interpersonal pressure. This finding may be explained in the Chinese cultural context. In China, the genders have different role expectations. Compared to females, males are expected to undertake more responsibilities and pressures from family and society. Therefore, boys have more experience and strategies than girls to deal with NLEs, such as conflicting with family members and friends, which in turn leads to suicidal ideation.” in the manuscript, page 8, lines 11-17.
“Possible explanations for this association are that individuals with very high self-esteem may consider that they are incapable of failing and indicative of perfectionist, a concept more readily linked to suicide thoughts. When encountering NLEs, people with high levels of self-esteem often cannot accept their own failures, and lack of effective response experience may be more likely to produce suicidal ideation.” in the manuscript, page 7, lines 36-38, and page 8, lines 1-2.
Comments: Finally, the authors should more deeply discuss what are, according to their expertise, the most relevant future interventions in order to reduce SI and suicide attempts in adolescents. Here, more details and some conclusive remarks about this topic are needed.
Response: Thanks for your suggestion. We have added the statements - “For example, to develop appropriate self-esteem with mindfulness training in the high-risk adolescent population, such as body scan, breath meditation, and emotion and thought meditation, could prevent an expected increase in SI. Beyond education and practical assistance, peer mentoring or support groups may be effective in reducing vulnerability of students to negative life events by increasing social support. ” in the manuscript, page 8, lines 39-44.
Round 2
Reviewer 1 Report
The paper has been revised to a satisfactory standard.
Reviewer 2 Report
In the revised version of the paper, the authors addressed significantly most of the major questions raised by Reviewers. I have no further additional comments.